# MiR-196a Promotes Lipid Deposition in Goat Intramuscular Preadipocytes by Targeting *MAP3K1* and Activating PI3K-Akt Pathway

**DOI:** 10.3390/cells13171459

**Published:** 2024-08-30

**Authors:** Yuling Yang, Wenyang Zhang, Haiyang Li, Hua Xiang, Changhui Zhang, Zhanyu Du, Lian Huang, Jiangjiang Zhu

**Affiliations:** 1Qinghai-Tibetan Plateau Animal Genetic Resource Reservation and Utilization Key Laboratory of Sichuan Province, Southwest Minzu University, Chengdu 610041, China; yulingyang1012@163.com (Y.Y.); haiyang6715@163.com (H.L.); xianghua2008411@163.com (H.X.); zhangchanghui0074@163.com (C.Z.); yuzhan.du@outlook.com (Z.D.); 2Key Laboratory of Qinghai-Tibetan Plateau Animal Genetic Resource Reservation and Utilization, Southwest Minzu University, Ministry of Education, Chengdu 610041, China; zhangwenyang6588@163.com

**Keywords:** miR-196a, MAP3K1, PI3K-Akt pathway, IMF, RNA-seq

## Abstract

Meat quality in goats is partly determined by the intramuscular fat (IMF) content, which is associated with the proliferation and differentiation of intramuscular preadipocytes. Emerging studies have suggested that miRNA plays a crucial role in adipocyte proliferation and differentiation. In our recent study, we observed the expression variations in miR-196a in the longissimus dorsi muscle of Jianzhou goats at different ages. However, the specific function and underlying mechanism of miR-196a in IMF deposition are still unclear. This study demonstrated that miR-196a significantly enhanced adipogenesis and apoptosis and reduced the proliferation of preadipocytes. Subsequently, RNA-seq was employed to determine genes regulated by miR-196a, and 677 differentially expressed genes were detected after miR-196a overexpression. The PI3K-Akt pathway was identified as activated in miR-196a regulating intramuscular adipogenesis via Kyoto Encyclopedia of Genes and Genomes (KEGG) analysis and further verified via Western blot and rescue assays. Lastly, using RT-qPCR, Western blot, dual-luciferase, and rescue assays, we found that miR-196a promoted adipogenesis and suppressed the proliferation of intramuscular preadipocytes by the downregulation of *MAP3K1*. In summary, these results suggest that miR-196a regulates IMF deposition by targeting *MAP3K1* and activating the PI3K-Akt pathway and provide a theoretical foundation for improving goat meat quality through molecular breeding.

## 1. Introduction

Goat meat is widely recognized as a high-quality meat that fully meets the healthy eating habits of modern society. It is well received by consumers because of its tender meat, unique flavor, and delicious taste. Intramuscular fat (IMF) content is one of the key factors that determine the tenderness and juiciness of meat and also greatly affects flavor [1,2]. The IMF content is affected by the differentiation and proliferation of intramuscular preadipocytes, which determine the adipocyte volume and number, respectively [3,4]. Research has demonstrated that the proliferation, differentiation, and apoptosis of preadipocytes are influenced by not only functional genes but also non-coding RNAs such as miRNA (microRNA) [5,6]. For example, miR-383-5p is involved in adipogenesis and lipid accumulation by targeting *RAD51AP1* [7]. MiR-33a inhibits the differentiation of bovine preadipocytes through the IRS2-Akt pathway [8]. Bta-miR-150, miR-378, miR-23b, and miR-25 are also involved in adipogenesis and lipid metabolism [9,10,11,12]. Therefore, screening the key miRNA and elucidating its underlying molecular mechanism in intramuscular fat deposition in goats holds great theoretical and practical significance.

MiRNA is a kind of small non-coding RNA with a length of about 22 nt. Evidence suggests that miR-196a could mediate adipogenesis. For example, miR-196a could induce the differentiation of 3T3-L1 and porcine preadipocytes [13,14]. The miR-196a-*Hoxc8*-*C/EBPβ* signaling pathway may be a therapeutic target by inducing brown adipogenesis to combat obesity and type 2 diabetes [15]. Additionally, miR-196a is highly expressed in visceral fat [16]. In our previous study [17], differences in the expression of miR-196a were observed in the longissimus dorsi muscle (LM) of Jianzhou goats at 2 months, 9 months, and 24 months old. Therefore, it is particularly important to clarify the function and underlying mechanism of miR-196a in goat IMF deposition.

Lipid metabolism in animals is regulated by a series of lipid metabolism signaling pathways [18]. PI3K-Akt is a key pathway regulating lipid metabolism in hepatic and diabetic nephropathy and affects the proliferation of porcine intramuscular preadipocytes [19,20,21]. Various miRNAs have also been shown to mediate adipogenesis through the PI3K-Akt pathway [22,23]. MAP3K1 (mitogen-activated protein kinase kinase kinase 1) is a serine and threonine kinase. *MAP3K1* can regulate insulin resistance as a candidate pathogenic gene in human preadipocytes and/or adipocytes [24]. Furthermore, *MAP3K1* can regulate lipid accumulation in chicken primary embryonic hepatocytes and affect the percentage of milk fat in Sahiwal cattle [25,26].

The results of this study show that miR-196a could promote adipogenesis and apoptosis and inhibit cell proliferation in goat intramuscular preadipocytes. Then, through RNA-seq and in vitro verification, we found that miR-196a regulated intramuscular preadipocyte adipogenesis by activating the PI3K-Akt signaling pathway. Furthermore, miR-196a was found to directly target *MAP3K1*, which was closely related to adipogenesis, to facilitate lipid deposition in preadipocytes. In summary, these data clarify that miR-196a can control lipid formation by targeting *MAP3K1* and activating the PI3K-Akt signaling pathway and provide the experimental basis for delineating the regulatory network of miR-196a in regulating goat IMF content.

## 2. Materials and Methods

### 2.1. Cell Isolation and Culture

Goat intramuscular preadipocytes were obtained from the longissimus dorsi muscle of three 2-day-old male Jianzhou goats. The methods of cell isolation and culture have been described above [27,28]. Briefly, the longissimus dorsi muscle tissues of three two-day-old goats were collected in a sterile environment, and equal amounts were mixed. After washing with PBS three times, it was broken on an ultra-clean bench. A two-fold volume of collagenase type II (Sigma-Aldrich, C2-BIOC, St. Louis, MO, USA) was used for digestion in a 37 °C water bath for 1.5 h, followed by the addition of an equal volume of DMEM/F12 (Gibco, Beijing, China) containing 10% fetal bovine serum (FBS) (TransSerum, Beijing, China) to finish digestion. The suspension was filtered by a 75 μm cell filter and centrifuged at 2000 rpm/min for 5 min. After adding red blood cell lysate (BOSTER, AR1118, Wuhan, China) and reaction for 5 min, the supernatant was discarded after centrifugation at 2000 rpm/min for 5 min. The cells were resuspended in DMEM/F12 (Gibco, Beijing, China) containing 10% FBS and seeded into a 25 cm^2^ cell culture flask and cultured at 37 °C and in 5% CO_2_. After 2 h, the medium (rich in muscle cells and other non-adherent cells) was discarded, the fresh medium was replaced, and then the medium was replaced every 2 days.

### 2.2. Cell Induction, siRNA Synthesis, and Cell Transfection

The goat intramuscular preadipocytes reached 80% confluence and were adipogenically induced by DMEM/F12 containing 10% FBS and 50 μM oleic acid (Sigma, St. Louis, MO, USA) as described in [29]. Mimics and inhibitors were synthesized according to the mature sequence of miR-196a by Guangzhou Ruibo Biotechnology Co., Ltd. (Guangzhou, China). The siRNA targeting the goat *MAP3K1* gene was designed and synthesized by Shanghai GenePharma Co., Ltd. (Shanghai, China), si-MAP3K1-1 (S: GCCCAGAAGAACGAAUGAUTT; AS: AUCAUUCGUUCUUCUGGGCTT); si-MAP3K1-2 (S: GGCUCUGCUAUUGGCAAAUTT; AS: AUUUGCCAAUAGCAGAGCCTT). The negative control si-NC was provided by GenePharma (S: UUCUCCGAACGUGUCACGUTT; AS: ACGUGACACGUUCGGAGAATT). When cell confluence reached 80%, Lipofectamine™3000 transfection reaction (Invitrogen, Carlsbad, CA, USA) was used for transfection according to the manufacturer’s instructions.

### 2.3. Oil Red O, Bodipy Staining, and Triglyceride Content Determination

According to a previous study [30], the cells were washed with PBS 3 times and fixed with 4% formaldehyde for 30 min. After removing the formaldehyde, the cells were washed with PBS again 3 times, and the Oil Red O dye solution (Solarbio, G1262, Beijing, China) was added. After standing at room temperature for 30 min, the dye solution was discarded. After PBS washing 3–5 times, images were captured using a microscope. Then, 600 μL of isopropanol was added to the 6-well plate and quantified by a microplate reader at a 510 nm wavelength. Similarly, 300 μL of 1 μg/mL Bodipy working solution was added to each well after the cells were fixed in formaldehyde, and the cells were incubated on a shaker at room temperature for 30 min. The Bodipy staining solution was discarded. After washing with PBS 3 times, 300 μL of 1 μg/mL DAPI staining solution was added to each well and incubated at room temperature for 10 min on a shaker. After PBS washing three times, a fluorescence microscope was used to take photos. The fluorescence intensity and cell number were measured by Image-Pro Plus 6.0 for quantitative analysis. Total triglycerides were extracted using a triglyceride detection kit (Applygen, E1013, Beijing, China), and the absorbance value was measured at a wavelength of 550 nm by a microplate reader. At the same time, a BCA protein quantification kit (Vazyme, Nanjing, China) was used for protein quantification, and the absorbance value was measured at a wavelength of 562 nm.

### 2.4. CCK-8 Assay

Cells were seeded into 96-well plates, and a miR-196a mimic, inhibitor, and the corresponding negative controls were transfected into the cells with a transfection reagent. After 0, 12, 24, 36, and 48 h of culture, 10 μL of CCK-8 (Life-ilab, Shanghai, China) solution was added to each well, and the absorbance at 450 nm was detected by a microplate reader after incubation at 37 °C for 30 min.

### 2.5. Analysis by Flow Cytometry

Cells were digested with trypsin without EDTA and collected. Propidium iodide (C0080, Solarbio, Beijing, China) and an Annexin V-FITC/PI apoptosis detection kit (A211, Vazyme, Nanjing, China) were used to detect the cell cycle classification and the proportion of apoptotic cells by flow cytometry according to the manufacturer’s instructions.

### 2.6. RNA Extraction and Reverse Transcription–Quantitative PCR

Total RNA was extracted using RNAiso Plus (Takara, 9109, Beijing, China). Then, 1 μg of total RNA was reverse-transcribed into cDNA using a reverse transcription kit (Vazyme, R32301, Nanjing, China). Then, a Taq Pro Universal SYBR qPCR Master Mix (Vazyme, Q712-02, Nanjing, China) was used for real-time fluorescence quantitative PCR detection by the Bio-Rad CFX96 PCR system. The reverse transcription of miRNA was performed by the stem–loop method, and U6 and UXT were used as internal controls for miR-196a and lipid metabolism-related gene detection, respectively. RT-qPCR was performed using gene-specific primers (Table 1). The relative expression was calculated by the 2^−∆∆CT^ method. The primer sequences are shown in Table 1.

### 2.7. RNA Sequencing (RNA-Seq)

RNA sequencing analysis was performed on intramuscular preadipocytes transfected with a mimic NC and a miR-196a mimic. After the RNA samples were collected, the cDNA library was constructed and sequenced by Lc-Bio Technologies (Hangzhou) Co., Ltd. (Hangzhou, China) DESeq2 screened the differential genes of the two groups, and the screening conditions were set at *p* < 0.05. KEGG was used to analyze the signaling pathway enrichment of the differentiated expressed genes.

### 2.8. Dual-Luciferase Reporter Assay

Primer Premier 5 was used to design the PCR primers for amplifying *MAP3K1* 3′-UTR. Wild-type *MAP3K1* (WT) was obtained by amplifying the gene fragment from cDNA. The mutant-*MAP3K1* (MUT) was obtained by overlap PCR. Then, wild-type and mutant pGL3-basic vectors were constructed. A miR-196a mimic or a mimic NC and WT or MUT dual-luciferase reporter vectors were co-transfected into goat intramuscular preadipocytes in a 6-well plate. At the same time, pRL-TK was transfected into the cells for normalization. After 48 h of transfection, the luciferase activity was measured using a dual-luciferase kit (Promega, Madison, WI, USA).

### 2.9. Western Blot

Cells were collected and lysed with cell lysis buffer for Western blotting (Beyotime, P0013, Shanghai, China), and appropriate protease inhibitors and phosphatase inhibitors were added at the same time. A BCA protein quantification kit (Thermo Fisher Scientific, 23225, Beijing, China) was used to detect the protein concentration. The protein was separated by an SDS-PAGE electrophoresis system and transferred to a PVDF membrane. The primary antibodies were as follows: anti-*β*-actin (1:8000, BM0627, BOSTER, Wuhan, China), anti-Akt1 (1:1000, ab32505; Abcam, Cambridge, ENG, UK), anti-p-Akt1 Ser473 (1:2000, 4060; Cell Signaling Technology, Danvers, MA, USA), anti-p-FAK (1:1000, ab81298, Abcam, Cambridge, ENG, UK), anti-FAK (1:1000, 3285, Cell Signaling Technology, Danvers, MA, USA), anti-MAP3K1 (1:500, bs-18780R; Bioss, Beijing, China), and anti-caspase3 (1:1000, WL02117, Wanleibio, Shenyang, China). Goat anti-mouse IgG coupled to horse radish peroxidase (HRP; 1:1000, BA1050; BOSTER, Wuhan, China) and goat anti-rabbit IgG coupled to HRP (1:8000, BA1054; BOSTER, Wuhan, China) were used as secondary antibodies. Image J (1.51j8, NIH, Bethesda, MD, USA) was used to calculate the protein gray value.

### 2.10. Statistical Analysis

All quantitative data were expressed as means ± SEM. Analysis of variance was performed using the GraphPad Prism software (8.3.0, GraphPad Software, La Jolla, CA, USA). Student’s *t*-test and one-way ANOVA were used for statistical analysis between the two groups and multiple groups. The difference was statistically significant when *p* < 0.05.

## 3. Results

### 3.1. miR-196a Is Associated with Intramuscular Fat Deposition

Our previous RNA-seq data [17] revealed that miR-196a expression was downregulated in 9-month-old LM compared with 2-month-old LM, whereas it was upregulated in 24-month-old LM tissue (Figure 1A,B). To detect whether the intramuscular fat deposition from 9-month-old to 24-month-old LM tissues [31] was associated with miR-196a, we further explored the effect of miR-196a on intramuscular lipid formation. Firstly, we examined the expression of miR-196a during the differentiation of goat intramuscular preadipocytes. miR-196a expression gradually increased from day 0 to day 8 (Figure 1C). These results predict that miR-196a may play a crucial role in the adipogenesis of intramuscular preadipocytes.

### 3.2. Overexpression of miR-196a Promotes Adipogenesis and Inhibits Proliferation of Goat Intramuscular Preadipocytes

To verify the function of miR-196a in intramuscular preadipocyte adipogenesis, we transfected a synthetic miR-196a mimic into goat primary intramuscular preadipocytes. As shown in Figure 2A, the expression of miR-196a was significantly increased after miR-196a mimic transfection. Then, we examined the content of triglyceride (TAG), lipid droplets, and the expression of lipid metabolism-related genes after the overexpression of miR-196a. The results show that the triglyceride content was significantly increased (Figure 2B). Correspondingly, Oil Red O and Bodipy staining showed that the overexpression of miR-196a increased the lipid droplet content (Figure 2C and Appendix A). The RT-qPCR results demonstrate that the overexpression of miR-196a increased the expression of fatty acid desaturase gene *SCD1* and fatty acid elongation gene *ELOVL6*. Additionally, there was an apparent increase in the expression of lipid synthesis genes *GPAM* and *DGAT1*, while the expression of *DGAT2* was decreased. Furthermore, the lipolysis genes *LPL* and *ACOX1* showed a significant increase after miR-196a overexpression, and the expression of *ATGL* was downregulated. When compared with the mimic NC group, the overexpression of miR-196a resulted in an expression increase in transcription regulators *PPARα*, *C/EBPα*, and *C/EBPβ* and an abundance decrease in *SREBP1c* (Figure 2D). These data reveal that miR-196a promotes the adipogenesis of goat intramuscular preadipocytes.

Next, we studied the effect of miR-196a on the proliferation and apoptosis of intramuscular preadipocytes. The CCK-8 results show that miR-196a overexpression inhibited cell proliferation (Figure 2E). In addition, cell cycle analysis showed that overexpression of miR-196a increased the proportion of G0/G1-phase cells and decreased the proportion of cells in the S and G2/M phases (Appendix A). At the same time, the expressions of proliferation marker genes cyclin-dependent-kinase 1 (*CDK1*) and proliferating cell nuclear antigen (*PCNA*) were downregulated after the overexpression of miR-196a (Appendix A). Flow cytometry showed that the apoptosis percentage was increased after miR-196a overexpression (Figure 2F). RT-qPCR also exhibited the increased expression of *caspase3* and decreased expression of *BCL2* (Figure 2G). Correspondingly, the WB results show that caspase3 protein expression increased after the miR-196a mimic treatment (Figure 2H). The above data indicate that miR-196a inhibits the proliferation and facilitates the apoptosis of goat intramuscular preadipocytes.

### 3.3. Knockdown of miR-196a Inhibits Adipogenesis and Promotes Proliferation of Goat Intramuscular Preadipocytes

Then, we knocked down the expression of miR-196a, and the intracellular triglyceride content was decreased after the transfection of the miR-196a inhibitor (Figure 2A,B). Oil Red O and Bodipy staining showed that the lipid droplet content was reduced (Figure 3C and Appendix A). Correspondingly, the abundances of the fatty acid desaturation gene *SCD1*, the lipid synthesis genes *GPAM* and *DGAT1*, the lipolysis genes *LPL* and *ACOX1*, and transcriptional regulators (*PPARα*, *PPARγ*, *C/EBPα*, and *C/EBPβ*) all declined, while the expressions of the lipid synthesis gene *DGAT2* and the lipolysis gene *ATGL* were upregulated (Figure 2D). Overall, our results suggest that interference with miR-196a hinders the adipogenesis of goat intramuscular preadipocytes.

To investigate the impact of interfering with miR-196a on the proliferation and apoptosis of goat intramuscular preadipocytes, we carried out the CCK-8 assay on the miR-196a inhibitor and NC cells. Cell viability was increased after interference with miR-196a (Figure 3E). Cell cycle analysis showed that interference with miR-196a reduced the proportion of G0/G1-phase cells and increased the proportion of S and G2/M phases cells (Appendix A). RT-qPCR analysis revealed a notable upregulation in the expression of proliferation-related genes *CDK4* and *CDK1* (Appendix A). Furthermore, apoptosis analysis revealed that treatment with the miR-196a inhibitor exhibited a lower apoptotic rate compared with the inhibitor NC group. This was accompanied by a decrease in the expression of the apoptosis-related genes *Bax*, *caspase3*, and *caspase7*, while the expression of *BCL2* was upregulated (Figure 3F,G). Correspondingly, the WB results show that caspase3 protein expression was decreased after miR-196a inhibitor treatment (Figure 3H). In summary, interference with miR-196a can inhibit the adipogenesis of goat intramuscular preadipocytes and promote their proliferation.

### 3.4. Identification and Analysis of Differentially Expressed Genes after miR-196a Overexpression

To investigate the underlying molecular mechanism by which miR-196a promotes lipid deposition, we submitted the total RNAs of the miR-196a mimic and mimic NC groups for transcriptome sequencing. A total of 677 differentially expressed genes (DEGs) (294 downregulated and 383 upregulated) were identified (Figure 4A,B; Appendix A). KEGG analysis revealed that DEGs were involved in multiple signaling pathways, including several lipid metabolism-related signaling pathways, focal adhesion, and the PI3K-Akt pathway (Figure 4C; Appendix A).

### 3.5. miR-196a Promotes Lipid Accumulation by Activating PI3K-Akt Pathway

It has been reported that focal adhesion and PI3K-Akt signaling pathways are involved in lipid deposition [32,33]. Based on the above sequencing data, miR-196a could activate focal adhesion and PI3K-Akt signaling pathways; therefore, we hypothesized that miR-196a promoted adipogenesis through the regulation of focal adhesion and PI3K-Akt signaling pathways. In order to verify our hypothesis, we investigated the protein levels of FAK and Akt kinases in their active/phosphorylated and total forms in preadipocytes treated with the miR-196a mimic and mimic NC. The ratio of p-FAK/FAK did not change after the overexpression of miR-196a (Appendix A). Of note, the level of p-Akt and the ratio of p-Akt/Akt were increased after the overexpression of miR-196a (Figure 5A,C). Subsequently, we used LY294002 (a specific PI3K inhibitor (30 nM), later represented as PI3K-) to inhibit the PI3K-Akt signaling and detected a significant decrease in Akt phosphorylation and the p-Akt/Akt ratio (Figure 5B,D). After that, we performed a rescue experiment and detected the lipid droplet content. The results show that the lipid droplet accumulation induced by miR-196a was rescued via the inhibition of the PI3K-Akt signaling (Figure 5E,F). These results reveal that miR-196a induces adipogenesis by targeting the PI3K-Akt signaling pathway in goat intramuscular preadipocytes.

### 3.6. MAP3K1 Is a Direct Target Gene of miR-196a in Goat Intramuscular Preadipocytes

To find the direct target gene of miR-196a in goat intramuscular preadipocytes, we analyzed the bioinformatic databases (miRWalk and TargetScan) and our transcriptome data and found that *MAP3K1* might be a target gene of miR-196a (Figure 6A). To verify this hypothesis, we detected the mRNA expression of *MAP3K1* after the overexpression and knockdown of miR-196a, respectively. The RT-qPCR results show that the miR-196a mimic reduced the expression of *MAP3K1*, while the miR-196a inhibitor facilitated *MAP3K1* expression (Figure 6B,C). On this basis, we performed a dual-luciferase reporter gene assay. The results show that miR-196a inhibited wild-type *MAP3K1*-3′-UTR luciferase activity, and this inhibition was completely eliminated when the binding site in the 3′-UTR of *MAP3K1* was mutated (Figure 6D). Moreover, we detected the MAP3K1 protein level and found that the overexpression of miR-196a reduced MAP3K1 protein expression, while the knockdown of miR-196a increased the protein expression of MAP3K1 (Figure 6E). The above results indicate that miR-196a directly targets *MAP3K1* by binding to its 3′-UTR sequence.

### 3.7. miR-196a Promotes Adipogenesis and Inhibits Proliferation by Targeting MAP3K1 in Goat Intramuscular Preadipocytes

To further confirm the role of *MAP3K1* in miR-196a-regulated lipid accumulation, we synthesized two pairs of si-*MAP3K1* and detected their interference efficiency. The interference efficiency of si-*MAP3K1*-1 was higher, so it was selected for subsequent experiments (Appendix A). The protein abundance of MAP3K1 was decreased after the transfection of si-*MAP3K1*-1 by WB (Figure 7A). As expected, Bodipy and Oil Red O staining revealed that the lipid droplet content was increased in the si-*MAP3K1* group compared with the si-NC group (Figure 7C). The RT-qPCR results exhibit the increased expression of the lipid metabolism genes *ACSL1*, *DGAT1*, *DGAT2,* and *SREBP1c* and the decreased expression of *LPL* after interference with *MAP3K1* (Figure 7B). The CCK-8 results show that the suppression of *MAP3K1* inhibited cell proliferation and the expression of proliferation marker genes *PCNA* and *CDK1* (Figure 7D and Appendix A). Subsequently, we carried out a rescue assay by interfering with *MAP3K1* in miR-196a inhibition preadipocytes and found that interfering with *MAP3K1* reversed the inhibitory effect on the lipid droplet content induced by miR-196a suppression via Oil Red O staining (Figure 7E). In conclusion, it can be concluded that miR-196a promotes lipid deposition and inhibits proliferation via directly targeting *MAP3K1* in goat intramuscular preadipocytes.

## 4. Discussion

IMF content is a key factor affecting the quality of goat meat, and IMF deposition is associated with intramuscular preadipocyte proliferation and differentiation [34]. In recent years, miRNAs have been described to be involved in adipogenesis and lipid metabolism by regulating adipocyte proliferation, differentiation, and apoptosis [35]. miR-196a has been extensively studied for its vital role in cancer and disease treatment [36,37]. Meanwhile, it has also been found that miR-196a has multiple effects on adipogenesis [13,15]. It has been reported that miR-196a can induce the differentiation of porcine preadipocytes, and compared with piglets, the expression of miR-196a was significantly upregulated in the adipose tissue of adult pigs [13]. Our previous study exhibited that, compared with 9-month-old Jianzhou goats, the expression of miR-196a in the longissimus dorsi of 24-month-old Jianzhou goats was upregulated by 3.59 times, corresponding to its increased IMF content displayed in our other study [31]. Consistent with these findings in longissimus dorsi tissue, we found that the expression of miR-196a was progressively elevated during days 0–8 of adipogenic differentiation in goat intramuscular preadipocytes. Similarly, the abundance of miR-196a was also upregulated during human preadipocyte differentiation [38]. These findings predict that miR-196a may play a positive role in goat intramuscular preadipocyte adipogenesis.

*SCD1* catalyzes the production of oleoyl-CoA (18:1) and palmitoyl-CoA (16:1), which are the main components of TAG [39,40]. Our work showed that miR-196a promoted the expression of *SCD1* in intramuscular preadipocytes. Therefore, the accumulation of monounsaturated fatty acids, induced by *SCD1*, resulted in the formation of TAG and lipid droplets in the present study. *LPL* promotes the hydrolysis of TAG in the blood into fatty acids, which can be absorbed to participate in TAG synthesis in adipocytes [41,42]. *ATGL* is the rate-limiting enzyme in the first step of TAG hydrolysis [43]. Thus, miR-196a-promoted lipid deposition can be regulated by promoting fatty acid uptake and TAG synthesis. *PPARγ*, *C/EBPα,* and *C/EBPβ* are key transcription regulators that promote preadipocyte differentiation [44,45]. As a consequence, the downregulation of these transcription regulators after interfering with miR-196a contributes to reduced lipid accumulation.

In adipose tissue, the fat mass is determined by the adipocyte volume and number [7], which are mainly controlled by the proliferation and adipogenic differentiation of preadipocytes [46]. As expected, miR-196a-promoted lipid deposition is accompanied by cell proliferation inhibition and the cell apoptosis promotion of preadipocytes, which is consistent with miR-196a’s function in mice [47,48,49]. PCNA is known as a molecular marker of proliferation, and CDK1 and CDK4 are key cell cycle regulators [50,51,52]. Consistent with the function of these key enzymes, our results reveal the downregulation of *PCNA* and *CDK1* in the miR-196a mimic group, together with the upregulation of *CDK1* and *CDK4* in the miR-196a inhibitor group. Consistent with the inhibition of miR-196a in intramuscular preadipocyte proliferation, the cell cycle was arrested in the S and G2/M phases. Furthermore, apoptosis is an important factor affecting the number of cells. The apoptotic rate was increased after miR-196a overexpression in intramuscular preadipocytes, and the apoptosis marker genes *BCL2*, *Bax*, and *caspase3/7* were changed, correspondingly. Similar to the findings in goat intramuscular preadipocytes, the upregulation of miR-196a-5p leads to increased apoptosis in oral squamous cell carcinoma cells [53].

The PI3K-Akt pathway is closely related to adipogenesis and can control tumor development and lipid synthesis [54]. Activated Akt (p-Akt) can phosphorylate a series of substrates, thereby regulating a variety of cellular functions, such as cell survival, differentiation, apoptosis, and adipogenesis [55]. Notably, DEGs in miR-196a-overexpressing cells were enriched in the PI3K-Akt pathway, and the overexpression of miR-196a authentically promoted AKT phosphorylation. Consistently, miR-196a facilitated Akt phosphorylation in the human colorectal cancer cell line SW480 and activated the AKT signaling pathway [56]. Thus, we propose that miR-196a might promote lipid deposition through the PI3K-Akt pathway. LY294002 is a specific PI3K inhibitor that suppresses PI3K phosphorylation and then influences the activation of downstream Akt signaling [57]. Therefore, we used LY294002 to inhibit the PI3K-Akt signaling pathway in intramuscular preadipocytes and found that miR-196a-promoted lipid deposition in goat intramuscular preadipocytes was realized by activating Akt signaling. In Figure 4C, we observe that the herpes simplex virus 1 infection pathway is significantly enriched. Herpes simplex virus type 1 (HSV-1) and type 2 (HSV-2) are enveloped double-stranded DNA viruses belonging to Herpesviridae [58]. The PI3K-Akt signaling pathway is essential for cell survival and can be activated by various viruses such as HIV-1 and HSV-1 [59,60]. HSV-1 infection increased the expressions of P-PI3K, P-AKT, and p-p38 in HeLa and Vero cells [60]. Moreover, sophoridine (SRI) treatment significantly inhibited the HSV-1 infection-induced phosphorylation of PI3K, Akt, and mTOR in Vero and HeLa cells [61]. A total of 27 genes were enriched in the herpes simplex virus 1 infection pathway in the RNA-Seq data of this study. Among them, component 3 (C3) has a protective effect on delaying keratitis and neovascularization after HSV-1 infection [62]. HSV-1 susceptibility increased in the fibroblasts of *STAT1*-deficient patients [63]. Therefore, we speculate that miR-196a may promote HSV-1 infection by downregulating the expression of genes such as *C3* and *STAT1* and then activating the PI3K-Akt pathway. The specific molecular mechanisms involved need to be further studied. This discovery provides us with a new interesting idea.

Mitogen-activated protein kinase kinase kinase 1 (MAP3K1) is responsible for transmitting signals from MAP2K to MAPK cascades, thereby controlling cell survival, maturation, and development [64]. Moreover, *MAP3K1* plays a role in miR-375’s suppression of lipid droplets and triglyceride formation in chicken liver [25]. In the current study, we found that interference with *MAP3K1* facilitated lipogenesis and inhibited cell proliferation by affecting the expression of genes associated with lipid metabolism and proliferation. After that, rescue experiments showed that miR-196a promotes lipid deposition by targeting *MAP3K1*. Interestingly, a previous study has shown that the PI3K-Akt pathway may play a role in the effects elicited via MAPK signaling [65], and MAP3K1 plays a key role in MAPK signaling; correspondingly, we wondered whether *MAP3K1*, a target of miR-196a, could regulate lipid deposition in intramuscular preadipocytes through PI3K-Akt signaling and detected the p-Akt level after interfering with *MAP3K1*. Our results show that suppressing *MAP3K1* expression did not affect the phosphorylation of cellular Akt (Appendix AD).

In summary, the expression of miR-196a was upregulated during adipogenesis in vitro and in vivo. The current study revealed that miR-196a promotes adipogenesis and inhibits the cell proliferation of goat intramuscular preadipocytes. Moreover, our data suggest that miR-196a promotes lipid formation in intramuscular preadipocytes by targeting *MAP3K1* and activating the PI3K-Akt pathway. These findings provide an experimental and theoretical basis for improving the quality of goat meat by manipulating the abundance of miR-196a in intramuscular fat.

## 5. Conclusions

Our findings suggest that miR-196a is involved in the coordinated reprogramming of metabolism and cell function in intramuscular preadipocytes from proliferation to differentiation, promotes adipogenesis, and inhibits proliferation in goat intramuscular preadipocytes. This is achieved through the activation of the PI3K-Akt pathway and regulation of *MAP3K1*. These findings shed light on the involvement of miR-196a in IMF formation in skeletal muscle through the coordinated regulation of cell proliferation and adipogenesis and offer a theoretical foundation for improving the quality of goat meat through molecular breeding.

## Figures and Tables

**Figure 1 cells-13-01459-f001:**
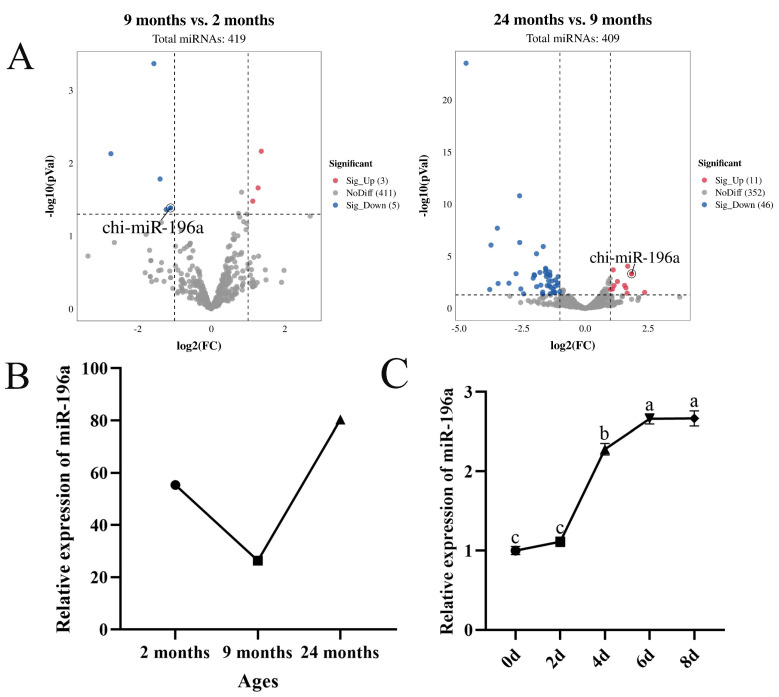
miR-196a is associated with intramuscular fat deposition. (**A**) Volcano plots of differentially expressed miRNAs in LM tissues of 9 months vs. 2 months and 24 months vs. 9 months. miR-196a is marked by circular ring. (**B**) The expression level of miR-196a in longissimus dorsi muscle at different developmental stages (2 months, 9 months, and 24 months) (n = 3). (**C**) Expression pattern of miR-196a during differentiation of goat preadipocytes. Different lowercase letters indicate significant differences (*p* < 0.05).

**Figure 2 cells-13-01459-f002:**
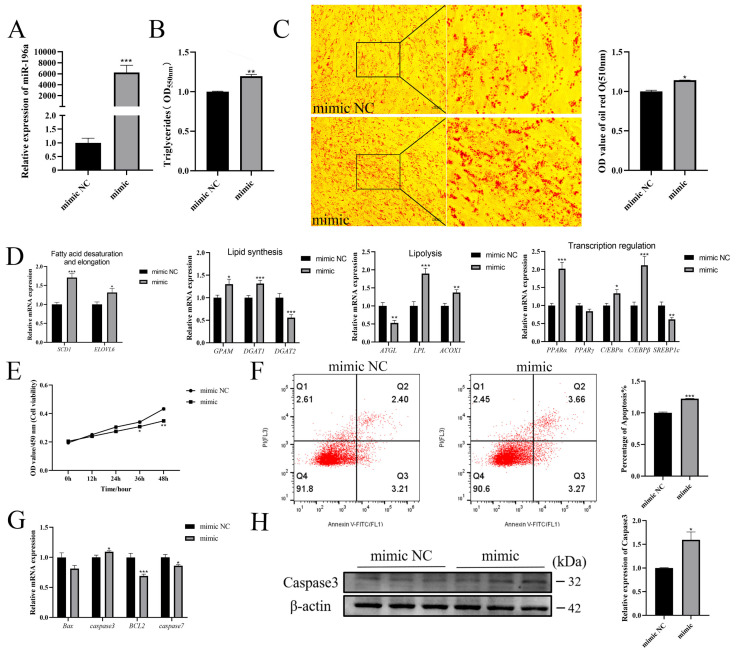
Overexpression of miR-196a promotes adipogenesis of goat intramuscular preadipocytes and inhibits their proliferation. (**A**) miR-196a overexpression efficiency detection. U6 was used as the internal reference gene and the mimic NC as the control group. (**B**) Triglyceride content detection after treatment with miR-196a mimic. (**C**) Oil Red O staining and quantification of lipid droplets differentiated for two days after overexpression of miR-196a (day 2 after adipogenic differentiation; 100×). (**D**) The relative expressions of lipid metabolism-related genes after miR-196a mimic treatment (day 2 after adipogenic differentiation). (**E**) The cell viability detection after overexpression of miR-196a. (**F**) Apoptosis examination after miR-196a mimic and corresponding NC treatment. (**G**) The relative expression level of apoptosis-related genes after overexpression of miR-196a. (**H**) Relative protein expression of caspase3 in cells after transfection with miR-196a mimic or mimic NC. The data are expressed as means ± SEM. * *p* < 0.05, ** *p* < 0.01, and *** *p* < 0.001.

**Figure 3 cells-13-01459-f003:**
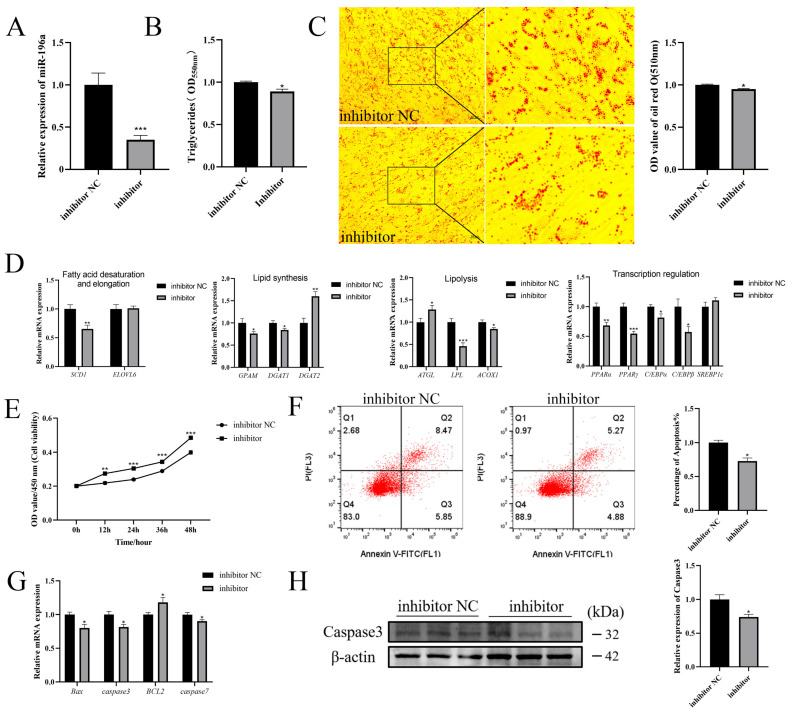
Knockdown of miR-196a inhibits adipogenesis of goat intramuscular preadipocytes and promotes their proliferation. (**A**) Detection of the interference efficiency after miR-196a inhibitor treatment. U6 was used as the internal reference gene. (**B**) Triglyceride content detection in miR-196a inhibitor and corresponding NC-treated cells. (**C**) Oil Red O staining and quantification of lipid droplets after interference with miR-196a (day 2 after adipogenic differentiation; 100×). (**D**) The relative expression of lipid metabolism-related genes after miR-196a inhibitor treatment (day 2 after adipogenic differentiation). (**E**) Cell viability detection after interfering with miR-196a. (**F**) Apoptosis detection after miR-196a inhibitor and corresponding NC treatment. (**G**) Relative expression of apoptosis-related genes after interfering with miR-196a. (**H**) Relative protein expression of caspase3 in cells after transfection with miR-196a inhibitor or inhibitor NC. The data are expressed as means ± SEM. * *p* < 0.05, ** *p* < 0.01, and *** *p* < 0.001.

**Figure 4 cells-13-01459-f004:**
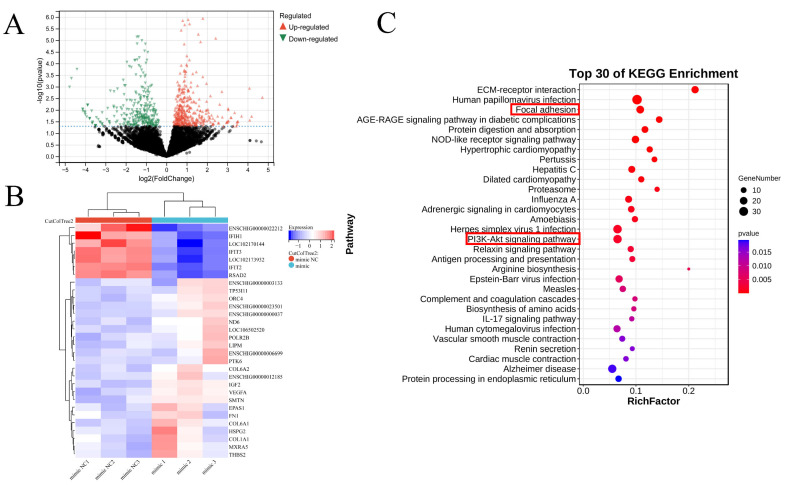
Analysis of differentially expressed genes in RNA-seq data of miR-196a-overexpressing and NC intramuscular preadipocytes. (**A**) Volcano map of DEGs in preadipocytes treated with miR-196a mimic and mimic NC. Red dot indicates significantly upregulated genes; green dot indicates significantly downregulated genes. (**B**) Heat map of the top 30 DEGs after overexpressing miR-196a in preadipocytes. Blue color–red color of band represents the expression upregulation of DEGs in miR-196a mimic and mimic NC groups. (**C**) The top 30 enriched pathways of DEGs after miR-196a overexpression in preadipocytes. The red boxes indicates that DEGs were involved in two lipid metabolism-related signaling pathways, focal adhesion, and the PI3K-Akt pathway.

**Figure 5 cells-13-01459-f005:**
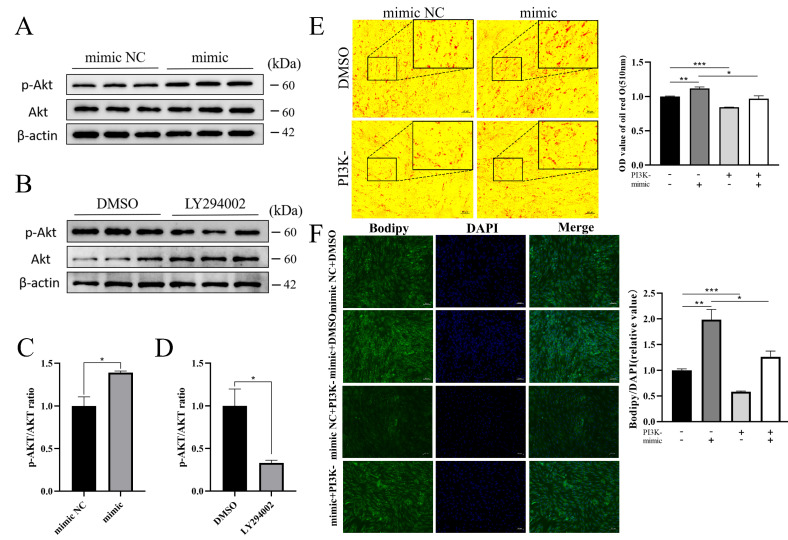
miR-196a promotes lipid accumulation in intramuscular preadipocytes by activating PI3K-Akt signaling pathway. (**A**,**C**) Detection of phospho-AKT (p-AKT-S473) and total AKT proteins abundances and ratio of p-Akt/Akt in miR-196a-overexpressing cells. (**B**,**D**) Detection of p-Akt and total Akt protein expression, and ratio of p-Akt/Akt in PI3K inhibitor (LY294002; PI3K)-treated and NC cells. (**E**) Lipid droplet content detection in miR-196a-overexpressing and control cells treated with PI3K inhibitor and NC by Oil Red O staining and quantification (day 2 after adipogenic differentiation; 200×). (**F**) Bodipy staining and quantification were performed for lipid droplet content examination in miR-196a-overexpressing and control cells treated with PI3K inhibitor and NC (day 2 after adipogenic differentiation; 100×). The data are expressed as means ± SEM. * *p* < 0.05, ** *p* < 0.01, and *** *p* < 0.001.

**Figure 6 cells-13-01459-f006:**
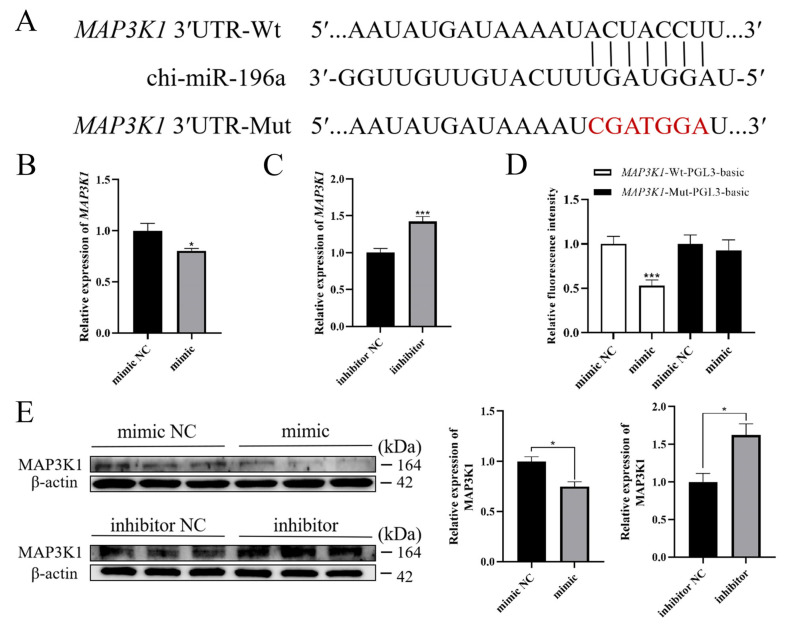
miR-196a regulates the expression of *MAP3K1* by binding to *MAP3K1* 3′-UTR. (**A**) The putative binding site sequence and mutated sequence of miR-196a in the 3′-UTR of *MAP3K1*. (**B**) The relative expression level of *MAP3K1* mRNA in cells transfected with miR-196a mimic and mimic NC. (**C**) The relative expression level of *MAP3K1* mRNA in cells transfected with miR-196a inhibitor and inhibitor NC. (**D**) Dual-luciferase assay in goat intramuscular preadipocytes co-transfected with miR-196a mimic or mimic NC and *MAP3K1*-Wt-PGL3-basic plasmid or *MAP3K1*-Mut-PGL3-basic vector. (**E**) The relative protein expression and quantification of MAP3K1 in cells transfected with miR-196a mimic, miR-196a inhibitor, or their corresponding NCs. The data are expressed as means ± SEM. * *p* < 0.05 and *** *p* < 0.001.

**Figure 7 cells-13-01459-f007:**
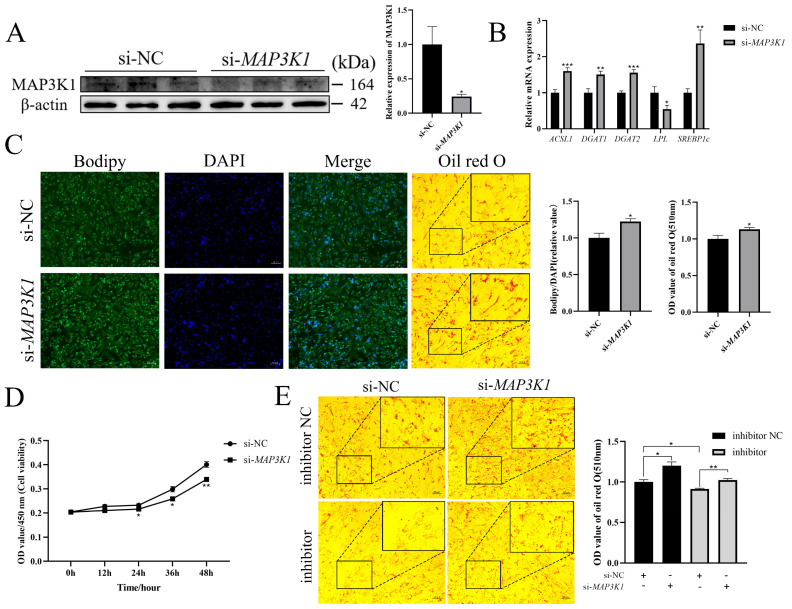
miR-196a promotes lipid accumulation in intramuscular preadipocytes by targeting *MAP3K1*. (**A**) Relative protein expression of MAP3K1 in cells after interference with *MAP3K1*. (**B**) Relative expression of lipid metabolism-related genes after si-*MAP3K1* treatment (day 2 after adipogenic differentiation). (**C**) Bodipy (100×) and Oil Red O (200×) staining and quantification of lipid droplets after interfering with *MAP3K1* (day 2 after adipogenic differentiation). (**D**) Detection of cell viability after interfering with *MAP3K1*. (**E**) Oil Red O staining and quantification of lipid droplets in miR-196a inhibition and control cells treated with si-*MAP3K1* and si-NC (day 2 after adipogenic differentiation; 200×). The data are expressed as means ± SEM. * *p* < 0.05, ** *p* < 0.01, and *** *p* < 0.001.

**Table 1 cells-13-01459-t001:** Primers for RT-qPCR analysis.

Gene	Full Name	Sequence	Tm/°C	Application
SCD1	Stearoyl-CoA desaturase 1	TGGCGTTCCAGAATGACGTT	60	RT-qPCR
		ACCCCATAGATACCACGGCA		
ELOVL6	Long-chain fatty acid family member 6	GGAAGCCTTTAGTGCTCTGGTC	60	RT-qPCR
		ATTGTATCTCCTAGTTCGGGTGC		
GPAM	Glycerol-3-phosphate acyltransferase, mitochondrial	GCAGGTTTATCCAGTATGGCATT	60	RT-qPCR
		GGACTGATATCTTCCTGATCATCTTG		
DGAT1	Diacylglycerol O-acyltransferase 1	CCACTGGGACCTGAGGTGTC	60	RT-qPCR
		GCATCACCACACACCAATTCA		
DGAT2	Diacylglycerol O-acyltransferase 2	CATGTACACATTCTGCACCGATT	60	RT-qPCR
		TGACCTCCTGCCACCTTTCT		
ATGL	Adipose triglyceride lipase	GGAGCTTATCCAGGCCAATG	60	RT-qPCR
		TGCGGGCAGATGTCACTCT		
LPL	Lipoprotein lipase	AGGACACTTGCCACCTCATTC	60	RT-qPCR
		TTGGAGTCTGGTTCCCTCTTGTA		
ACOX1	Acyl-CoA oxidase 1	CGAGTTCATTCTCAACAGTCCT	60	RT-qPCR
		GCATCTTCAAGTAGCCATTATCC		
PPARα	Peroxisome proliferator-activated receptor alpha	AGGTCCGCATCTTCCACT	60	RT-qPCR
		GCTTCGTAAACGCCATACTT		
PPARγ	Peroxisome proliferator-activated receptor gamma	AAGCGTCAGGGTTCCACTATG	60	RT-qPCR
		GAACCTGATGGCGTTATGAGAC		
C/EBPα	Enhancer-binding protein alpha	GCGGCAAAGCCAAGAAGTCC	60	RT-qPCR
		CGGCTCAGTTGTTCCACCC		
C/EBPβ	Enhancer-binding protein beta	GCCTGTCCACGTCCTCGTCGTCCAGC	60	RT-qPCR
		CGGATCTTGTACTCGTCGCTGTGCTTGTCC		
SREBP1c	Sterol regulatory element-binding protein 1c	ACGCCATCGAGAAACGCTAC	60	RT-qPCR
		GTGCGCAGACTCAGGTTCTC		
Bax	BCL2-associated X	TTTCCGACGGCAACTTCAA	60	RT-qPCR
		TGAGCACTCCAGCCACAAA		
Caspase3	Caspase3	GACGTGGATGCAGCAAACCTCA	60	RT-qPCR
		TTCACCATGGCTTAGAAGCACG		
Caspase7	Caspase7	GGAACAGATGGCAAGACAGCAATAAAG	60	RT-qPCR
		GCCTGAATGAAGAAGAGTTTGGGTTTC		
BCL2	Cell lymphoma-2	ATGTGTGTGGAGAGCGTCAA	60	RT-qPCR
		CCTTCAGAGACAGCCAGGAG		
PCNA	Proliferating cell nuclear antigen	ATCAGCTCAAGTGGCGTGAA	60	RT-qPCR
		TGCCAAGGTGTCCGCATTAT		
CDK4	Cyclin-dependent kinase 4	AAGTGGTGGGACAGTCAAGC	60	RT-qPCR
		ACAGAAGAGAGGCTTTCGACG		
CDK1	Cyclin-dependent kinase 1	AGATTTTGGCCTTGCCAGAG	60	RT-qPCR
		AGCTGACCCCAGCAATACTT		
ACSL1	Acyl-CoA synthetase long-chain family member 1	TGACTGTTGCTGGAGACTGG	60	RT-qPCR
		CAGCCGTCTTTATCCAGAGC		
MAP3K1	Mitogen-activated protein kinase kinase kinase 1	AGGCTCAAGATGTGGGAACTG	60	RT-qPCR
		ATCCTGATGATGTTTGGGTGAT		
chi-miR-196a	chi-miR-196a	TCGGCAGGTAGGTAGTTTCATG	60	RT-qPCR
		TGGAGTCGGCAATTCAGTTGA		
U6	U6	GCTTCGGCAGCACATATACTAAAAT	60	RT-qPCR
		CGCTTCACGAATTTGCGTGTCAT		

## Data Availability

The RNA-seq data have been deposited in the NCBI Sequence Read Archive (SRA), Bio project number PRJNA1013160.

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
