# Peer review of "MiR-196a Promotes Lipid Deposition in Goat Intramuscular Preadipocytes by Targeting MAP3K1 and Activating PI3K-Akt Pathway"

_cells, 2024, doi:10.3390/cells13171459_

Round 1

Reviewer 1 Report

Comments and Suggestions for Authors

The manuscript by Yuling Yang and coworkers, entitled: “MiR-196a Promotes Lipid Deposition in Goat Intramuscular Preadipocytes by Targeting MAP3K1 and Activating PI3K-Akt Pathway” describes an interesting study on a model of primary goat preadipocytes and adipocytes, showing that one of the microRNA molecules: mir-196a is involved in a positive regulation of adipogenesis, causing inhibition of MAP3K1 kinase expression, a decrease in preadipocytes proliferation, and enhancement of Akt kinase activity leading to stimulation of adipogenic differentiation. This manuscript presents results of a follow-up study of the Authors, who have already published several articles on the mechanism of differentiation of adipocytes forming the intramuscular fat in goat’s meat. In general, the study has been well planned, methods were chosen correctly and many results are interesting and convincing. However, I have some major concerns that affect the overall impression of the presented research.

Major remarks:

1.      The Authors don’t provide any information about the protocol of adipogenic differentiation of goat preadipocytes, although the study attempts to prove the role of  mir-196a in adipogenesis. The Materials and Methods section contains information about the method of cell isolation and culture, however, the Authors don’t inform about the method they used to separate muscle cells from preadipocytes. The reader is only informed about the elimination of erythrocytes by treating the cell pellet with red blood cell lysis buffer. Since this study is a continuation of this group’s previously published research I tried to look for this information in the previously published articles (Genes 2023, 14, 440. https://doi.org/10.3390/genes14020440; Int. J. Mol. Sci. 2023, 24, 13415. https://doi.org/10.3390/ijms241713415) but without any luck. In the article entitled: MCD Inhibits Lipid Deposition in Goat Intramuscular Preadipocytes, the Authors described induction of cell differentiation by treatment of preadipocytes with 50 μmol/L oleic acid for 1-6 days, but this is not a proper method of in vitro induction of adipogenic differentiation. It is crucial to provide this type of information to the reader, and to present convincing images of adipocytes after in vitro differentiation, especially when using such an uncommon model of goat cells. I know from personal experience that adjusting the method of adipogenic differentiation to cells isolated from species other then mouse or humans can be difficult, as the general protocol used for adipogenic differentiation of a well known model of 3T3-L1 preadipocytes doesn’t always work correctly.

2.      This leads to my second major remark regarding the quality of images presented in figures: 2C, 3C, 5E and F, 7C and E. The images of goat adipocytes stained with Oil red O or Bodipy are too small and taken from a very small magnification. Despite my true effort I was not able to see typical morphology of adipocytes on these images, even after enlargement of the manuscript’s file to 400%. In addition, the figure captions are missing information about the day of culture (day of adipogenic differentiation) at which the images were taken and the objective magnification that was used when the images were captured. It seems that the images were taken at a very low magnification, so it is hard to tell if they really present adipocytes or maybe myoblasts with a few adipoctyes scattered on a culture plate. The same doubts can be raised when reading the previously published papers that are also missing these information.

3.      The third major remark is connected with the use of siRNA in knock-down experiments. It is commonly known that the knock-down effect achieved after siRNA transfection usually is transient and becomes lost after  around 48-72h post-transfection. This means that it was easy to see the effect of MAP3K1 knock-down in preadipocytes (for example the effect on cell proliferation), but it could have been hard to observe it still in adipocytes after in vitro differentiation, which usually takes at least 4-6 days. Which brings us to the first question about the protocol used for adipogenic differentiation of goat preadipicytes and the day of culture at which the parameters of lipid accumulation or the expression of lipid metabolism-related genes were analyzed. This information is not provided in the figure captions.

4.      Western-blot images presented in fig. 5B don’t really correspond with the graph presented in fig.5D. Surprisingly, the Authors did not achieve inhibition of Akt kinase activity (based on its phosphorylation at Ser473)  after treatment with LY294002, although the bar graph shows a significant decrease of the p-Akt/Akt ratio.

Minor remarks:

1.      The manuscript needs some language corrections. It is generally well-written, but contains some grammar mistakes, and some sentences should be rephrased to improve their quality. Some examples are listed below, but they do not include all necessary corrections:

·        Abstract, lines: 19-20: RNA-seq was employed to depict the expression profiles in which 677 differentially expressed genes were captured after miR-196a overexpression. – This sentence should be rephrased, for example: RNA-seq was employed to determine genes regulated by  miR-196a, and 677 differentially expressed genes were detected after miR-196a overexpression.

·        Abstract, lines: 23-25: we found that miR-196a promoting adipogenesis and suppressing proliferation of intramuscular preadipocytes via targeting MAP3K1. This sentence should be rephrased, for example: … we found that miR-196a promoted adipogenesis and suppressed proliferation of intramuscular preadipocytes by downregulation of MAP3K1.

·        Introduction, line 31: Goat meat is widely recognized as a high-quality meat which fully meet the healthy eating habits of modern society. – grammar mistake (should be meets).

·        Introduction, line 51: In our previous study [17], significant differences were observed of the expression of miR-196a in the longissimus dorsi muscle(LM) – grammar mistake (should be differences in the expression).

·        Introduction, lines 68-71: – introduction should not provide conclusions.

·        Results, lines 260-263: A total of 677 differentially expressed genes (DEGs) (294 downregulated and 383 upregulated) were screened (Fig.4A-B, Table S2-S3). KEGG analysis revealed that DEGs were enriched in multiple signaling pathways, including several lipid metabolism-related signaling pathway focal adhesion and PI3K-Akt (Fig.4C, Table S4). This fragment should be rephrased, for example: A total of 677 differentially expressed genes (DEGs) (294 downregulated and 383 upregulated) were identified (Fig.4A-B, Table S2-S3). KEGG analysis revealed that DEGs were involved in multiple signaling pathways, including several lipid metabolism-related signaling pathway focal adhesion and PI3K-Akt pathway (Fig.4C, Table S4).

·        Results, lines275-278: … we wondered whether miR-196a promoting adipogenesis was through focal adhesion and PI3K-Akt pathways. In order to verify our inference, we detected the protein activating levels of FAK and Akt in preadipocytes treated with miR-196a mimic and mimic NC. This fragment should be rephrased, for example: …, therefore we hypothesized that miR-196a promoted adipogenesis through regulation of focal adhesion and PI3K-Akt signaling pathways. In order to verify our hypothesis, we investigated the protein levels of FAK and Akt kinases in their active/phosphorylated and total forms in preadipocytes treated with miR-196a mimic and mimic NC.

·        Discussion, line 354: Meanwhile, it has also been found to has multiple effect on adipogenesis.grammar mistake (should be: Meanwhile, it has also been found that miR-196a has multiple effects on adipogenesis.)

·        Discussion, line 367: Our work exhibited that miR-196a promoted … - grammar mistake (should be: Our work showed that miR-196a promoted…).

·        Discussion, lines 378: which are mainly controlled by proliferation and adipogenic of preadipocytes. – a word is missing, should be: which are mainly controlled by proliferation and adipogenic differentiation of preadipocytes.

·        Discussion, lines: 412-415: Intriguingly, previous study has showed that MAP3K1 could control the P-Ser473 Akt phosphorylation in mice, correspondingly, we wondered whether MAP3K1, target of miR-196a, regulating lipid deposition in intramuscular preadipocytes could through PI3K-Akt signaling, and detected the p-Akt level after interfering with MAP3K1. – citation is missing. Which previous study?

2.      The table presenting the primer sequences should be incorporated to the main text, not presented as supplementary table S1.

Comments on the Quality of English Language

The manuscript needs some language corrections. It is generally well-written, but contains some grammar mistakes, and some sentences should be rephrased to improve their quality. Some examples are listed in the main comments to the Authors (presented above). The manuscript should be corrected by a native speaker.

Author Response

Reply to the Review Report

Dear reviewer,

Thanks for your comments concerning our manuscript entitled “MiR-196a Promotes Lipid Deposition in Goat Intramuscular Preadipocytes by Targeting MAP3K1 and Activating PI3K-Akt Pathway” (cells-3114823). Those comments were valuable and very helpful. We have read the comments and the instructions for authors carefully. Revised portions are marked in yellow in the revised manuscript.

Comments and Suggestions for Authors

The manuscript by Yuling Yang and coworkers, entitled: “MiR-196a Promotes Lipid Deposition in Goat Intramuscular Preadipocytes by Targeting MAP3K1 and Activating PI3K-Akt Pathway” describes an interesting study on a model of primary goat preadipocytes and adipocytes, showing that one of the microRNA molecules: mir-196a is involved in a positive regulation of adipogenesis, causing inhibition of MAP3K1 kinase expression, a decrease in preadipocytes proliferation, and enhancement of Akt kinase activity leading to stimulation of adipogenic differentiation. This manuscript presents results of a follow-up study of the Authors, who have already published several articles on the mechanism of differentiation of adipocytes forming the intramuscular fat in goat’s meat. In general, the study has been well planned, methods were chosen correctly and many results are interesting and convincing. However, I have some major concerns that affect the overall impression of the presented research.

Major remarks:

  1. The Authors don’t provide any information about the protocol of adipogenic differentiation of goat preadipocytes, although the study attempts to prove the role of  mir-196a in adipogenesis. The Materials and Methods section contains information about the method of cell isolation and culture, however, the Authors don’t inform about the method they used to separate muscle cells from preadipocytes. The reader is only informed about the elimination of erythrocytes by treating the cell pellet with red blood cell lysis buffer. Since this study is a continuation of this group’s previously published research I tried to look for this information in the previously published articles (Genes2023, 14, 440. https://doi.org/10.3390/genes14020440; Int. J. Mol. Sci.2023, 24, 13415. https://doi.org/10.3390/ijms241713415) but without any luck. In the article entitled: MCD Inhibits Lipid Deposition in Goat Intramuscular Preadipocytes, the Authors described induction of cell differentiation by treatment of preadipocytes with 50 μmol/L oleic acid for 1-6 days, but this is not a proper method of in vitro induction of adipogenic differentiation. It is crucial to provide this type of information to the reader, and to present convincing images of adipocytes after in vitro differentiation, especially when using such an uncommon model of goat cells. I know from personal experience that adjusting the method of adipogenic differentiation to cells isolated from species other then mouse or humans can be difficult, as the general protocol used for adipogenic differentiation of a well known model of 3T3-L1 preadipocytes doesn’t always work correctly.

Response 1: Thank you for your comments. The main goal of our lab was to explore the cellular and molecular mechanism in adipogenesis of goat intramuscular preadipocytes. To achieve our objective, we have constructed the induction system of goat intramuscular preadipocyte-induction with 50 μM oleic acid for 48 h. Oleic acid, independent of insulin, promotes differentiation of goat primary preadipocytes in vitro. Oleic acid induction method has been established, published [1] and applied to the following studies [2, 3]. Of course, it has also been used in other species (chicken), oleate alone can act as a direct inducer of chicken preadipocyte differentiation by elevating expression of key positive regulators and suppressing expression of negative regulator of adipogenesis [4]. Compared with the traditional in vitro induction method, it is more convenient to use oleic acid alone to induce adipogenic differentiation, which effectively saves the induction time and can be used as an effective inducer of goat preadipocyte differentiation. We've added a description of cell adipogenic induction to the manuscript (lines: 90-92, highlighted).

Regarding your question “the Authors don’t inform about the method they used to separate muscle cells from preadipocytes”, we cited an articles [5] with more detailed method for isolating goat preadipocytes. At the same time, the description of cell isolation and culture methods in this paper was improved. In lines: 86-88, we added the method of isolating muscle cells from preadipocytes. According to our experience, preadipocytes are mostly adhered within 2 h, while muscle cells are still existed in the culture medium and can be isolated by changing the culture medium (highlighted).

Reference information:

  1. Tian, W.; Xiang, H.; Li, Q.; Wang, Y.; Zhu, J.; Lin, Y., Oleic acid, independent of insulin, promotes differentiation of goat primary preadipocytes. Animal Production Science 2023, 63, (2), 113-119.
  2. Li, X.; Zhang, H.; Wang, Y.; Li, Y.; Wang, Y.; Zhu, J.; Lin, Y., Chi-Circ_0006511 Positively Regulates the Differentiation of Goat Intramuscular Adipocytes via Novel-miR-87/CD36 Axis. International journal of molecular sciences 2022, 23, (20).
  3. Xiong, Y.; Xu, Q.; Lin, S.; Wang, Y.; Lin, Y.; Zhu, J., Knockdown of LXRα Inhibits Goat Intramuscular Preadipocyte Differentiation. International journal of molecular sciences 2018, 19, (10).
  4. Shang, Z.; Guo, L.; Wang, N.; Shi, H.; Wang, Y.; Li, H., Oleate promotes differentiation of chicken primary preadipocytes in vitro. Bioscience reports 2014, 34, (1).
  5. Xu, Q.; Wang, Y.; Zhu, J.; Zhao, Y.; Lin, Y., Molecular characterization of GTP binding protein overexpressed in skeletal muscle (GEM) and its role in promoting adipogenesis in goat intramuscular preadipocytes. Animal biotechnology 2020, 31, (1), 17-24.

  1. This leads to my second major remark regarding the quality of images presented in figures: 2C, 3C, 5E and F, 7C and E. The images of goat adipocytes stained with Oil red O or Bodipy are too small and taken from a very small magnification. Despite my true effort I was not able to see typical morphology of adipocytes on these images, even after enlargement of the manuscript’s file to 400%. In addition, the figure captions are missing information about the day of culture (day of adipogenic differentiation) at which the images were taken and the objective magnification that was used when the images were captured. It seems that the images were taken at a very low magnification, so it is hard to tell if they really present adipocytes or maybe myoblasts with a few adipoctyes scattered on a culture plate. The same doubts can be raised when reading the previously published papers that are also missing these information.

Response 2: Regarding the inability to see the morphology of adipocytes, we used a microscope model ZEISS Axio observer3, and in order to better show the content of lipid droplets (red), we used a yellow background for shooting, and we also tried to shoot a white background, but the effect was not satisfactory. Accordingly, the yellow background cannot clearly observe the cell morphology, and we will improve it in the future research by enlarging the magnification and adjusting the shooting method to obtain higher quality images.

About the question “the figure captions are missing information about the day of culture (day of adipogenic differentiation) at which the images were taken and the objective magnification that was used when the images were captured”,We have a ruler in the lower right corner of each picture, and about the induction time, we uniformly induce cell adipogenic differentiation for 2 days. A description has been added to the manuscript Fig.2C, 3C, 5E and F, 7C and E figure captions.

  1. The third major remark is connected with the use of siRNA in knock-down experiments. It is commonly known that the knock-down effect achieved after siRNA transfection usually is transient and becomes lost after around 48-72h post-transfection. This means that it was easy to see the effect of MAP3K1 knock-down in preadipocytes (for example the effect on cell proliferation), but it could have been hard to observe it still in adipocytes after in vitro differentiation, which usually takes at least 4-6 days. Which brings us to the first question about the protocol used for adipogenic differentiation of goat preadipicytes and the day of culture at which the parameters of lipid accumulation or the expression of lipid metabolism-related genes were analyzed. This information is not provided in the figure captions.

Response 3: We fully agree with your point of view, involving the first question, the protocol we used for goat preadipocyte differentiation was when goat intramuscular preadipocyte reached 80% confluence, and adipogenic differentiation was induced for 48 h with DMEM/F12 containing 10% FBS and 50 μM oleic acid (Sigma Corp.; Lewis, Missouri, USA). Therefore, knockdown assays can be performed using siRNA to observe the effect of knockdown of MAP3K1 on cell differentiation. we analyzed lipid accumulation parameters or lipid metabolism-related genes expression after 2 days of oleic acid-induced adipogenic differentiation, which has been described in the figure captions (highlighted).

  1. Western-blot images presented in fig. 5B don’t really correspond with the graph presented in fig.5D. Surprisingly, the Authors did not achieve inhibition of Akt kinase activity (based on its phosphorylation at Ser473) after treatment with LY294002, although the bar graph shows a significant decrease of the p-Akt/Akt ratio.

Response 4: In figure 5B, we can find that the expression of p-Akt was slightly but clearly decreased after LY294002 treatment, moreover, the expression of Akt was slightly increased after LY294002 treatment, so the p-Akt/Akt ratio in our bar chart of figure 5D showed a significant decrease.

Minor remarks:

  1. The manuscript needs some language corrections. It is generally well-written, but contains some grammar mistakes, and some sentences should be rephrased to improve their quality. Some examples are listed below, but they do not include all necessary corrections:

Abstract, lines: 19-20: RNA-seq was employed to depict the expression profiles in which 677 differentially expressed genes were captured after miR-196a overexpression. – This sentence should be rephrased, for example: RNA-seq was employed to determine genes regulated by miR-196a, and 677 differentially expressed genes were detected after miR-196a overexpression.

Abstract, lines: 23-25: we found that miR-196a promoting adipogenesis and suppressing proliferation of intramuscular preadipocytes via targeting MAP3K1. – This sentence should be rephrased, for example: … we found that miR-196a promoted adipogenesis and suppressed proliferation of intramuscular preadipocytes by downregulation of MAP3K1.

Introduction, line 31: Goat meat is widely recognized as a high-quality meat which fully meet the healthy eating habits of modern society. – grammar mistake (should be meets).

Introduction, line 51: In our previous study [17], significant differences were observed of the expression of miR-196a in the longissimus dorsi muscle (LM– grammar mistake (should be differences in the expression).

Results, lines 260-263: A total of 677 differentially expressed genes (DEGs) (294 downregulated and 383 upregulated) were screened (Fig.4A-B, Table S2-S3). KEGG analysis revealed that DEGs were enriched in multiple signaling pathways, including several lipid metabolism-related signaling pathway focal adhesion and PI3K-Akt (Fig.4C, Table S4). – This fragment should be rephrased, for example: A total of 677 differentially expressed genes (DEGs) (294 downregulated and 383 upregulated) were identified (Fig.4A-B, Table S2-S3). KEGG analysis revealed that DEGs were involved in multiple signaling pathways, including several lipid metabolism-related signaling pathway focal adhesion and PI3K-Akt pathway (Fig.4C, Table S4).

Results, lines275-278: … we wondered whether miR-196a promoting adipogenesis was through focal adhesion and PI3K-Akt pathways. In order to verify our inference, we detected the protein activating levels of FAK and Akt in preadipocytes treated with miR-196a mimic and mimic NC. – This fragment should be rephrased, for example: …, therefore we hypothesized that miR-196a promoted adipogenesis through regulation of focal adhesion and PI3K-Akt signaling pathwaysIn order to verify our hypothesis, we investigated the protein levels of FAK and Akt kinases in their active/phosphorylated and total forms in preadipocytes treated with miR-196a mimic and mimic NC.

Discussion, line 354: Meanwhile, it has also been found to has multiple effect on adipogenesis. – grammar mistake (should be: Meanwhile, it has also been found that miR-196a has multiple effects on adipogenesis.)

Discussion, line 367: Our work exhibited that miR-196a promoted … - grammar mistake (should be: Our work showed that miR-196a promoted…).

Discussion, lines 378: which are mainly controlled by proliferation and adipogenic of preadipocytes. – a word is missing, should be: which are mainly controlled by proliferation and adipogenic differentiation of preadipocytes.

Response: Thank you for your suggestions, we have corrected these grammatical problems in the manuscript (highlighted).

Introduction, lines 68-71: – introduction should not provide conclusions.

Response: We agree with you that we have deleted the conclusion in the introduction.

Discussion, lines: 412-415: Intriguingly, previous study has showed that MAP3K1 could control the P-Ser473 Akt phosphorylation in mice, correspondingly, we wondered whether MAP3K1, target of miR-196a, regulating lipid deposition in intramuscular preadipocytes could through PI3K-Akt signaling, and detected the p-Akt level after interfering with MAP3K1. – citation is missing. Which previous study?

Response: Thank you for your questions. Previous study has shown that the PI3K-Akt pathway may play a role in the effects elicited via MAPK signaling[6], and MAP3K1 plays a key role in MAPK signaling, correspondingly, we wondered whether MAP3K1, target of miR-196a, regulating lipid deposition in intramuscular preadipocytes could through PI3K-Akt signaling, and detected the p-Akt level after interfering with MAP3K1. We have discussed in the manuscript and added citations (line446-448, highlighted).

Reference information:

  1. Lee, J. T., Jr.; McCubrey, J. A., The Raf/MEK/ERK signal transduction cascade as a target for chemotherapeutic intervention in leukemia. Leukemia 2002, 16, (4), 486-507.

  1. The table presenting the primer sequences should be incorporated to the main text, not presented as supplementary table S1.

Response: We agree with you that we have incorporated the primer sequence table into the text, table 1.

We sincerely appreciate for your comments, and hope the responses will clarify the problems.

Best regards

Reviewer 2 Report

Comments and Suggestions for Authors

This study is well designed and performed to examine effects of miR-196a on goat preadipocyte differentiation, although there are several points to be addressed before publication.

Introduction: Why do authors focus on goat IMF preadipocytes? I have never heard of presence of goat IMF. Without any evidence of the presence, goat "preadipocyte" should be considered as a model of beef IMF cells.

Figure 1: Letters in Figure 1 are too small to understand. Please enlarge. Why do you think miR-196a expression declined from 2mo to 9 mo in LM muscle? Please discuss.

Figure 2: Letters in Figure 2 are too small, most of them are invisible. Please enlarge. In panel C, the background color should be gray and the black/white contrast should be stronger to understand the image.

Figure 3: Letters in the figure are too small, most of them are invisible. Please enlarge. In panel C, the background color should be gray and the black/white contrast should be stronger to understand the image.

Figure 4: Letters in the panels A and B are too small. Please enlarge.

Line 257-: Results from miR-196a inhibition by and  overexpression quite differed one another. Please discuss it.

Section 3.5: I do not agree to the conclusion that "miR-196a promotes lipid accumulation through activating PI3K-Akt pathway ". Authors examined effect of miR-196a by mimic and effect of LY294002 separately, but not in combination. If they want to conclude in above-mentioned way, they need to perform an examination using mimic and LY294002 in combining way.

Line 279: Why do authors think miR-196a altered overexpression p-Akt/Akt ratio but did not affect gene expression of adipogenic differentiation nor preadipocyte proliferation in RNA-seq? Please explain.

Figure 5: Letters in the panels C-F are too small, most of them are invisible. Please enlarge. In panel E, the background color should be gray and the black/white contrast should be stronger to understand the image. This issue is also true to Figures 6 and 7, needs to be addressed.

Author Response

Reply to the Review Report

Dear reviewer,

Thanks for your comments concerning our manuscript entitled “MiR-196a Promotes Lipid Deposition in Goat Intramuscular Preadipocytes by Targeting MAP3K1 and Activating PI3K-Akt Pathway” (cells-3114823). Those comments were valuable and very helpful. We have read the comments and the instructions for authors carefully. Revised portions are marked in yellow in the revised manuscript.

Comments and Suggestions for Authors

This study is well designed and performed to examine effects of miR-196a on goat preadipocyte differentiation, although there are several points to be addressed before publication.

Introduction: Why do authors focus on goat IMF preadipocytes? I have never heard of presence of goat IMF. Without any evidence of the presence, goat "preadipocyte" should be considered as a model of beef IMF cells.

Response1: Thank you for your question. It is well known that Intramuscular fat (IMF) is considered to be one of the most important factors affecting meat quality. Its content is positively correlated with meat tenderness, flavor and juiciness. In order to improve goat meat quality and IMF content through resolving the cellular and molecular mechanism of adipogenesis, we pay attention to goat intramuscular preadipocytes. In recent years, there have been many studies towards goat IMF and goat intramuscular preadipocytes [1-3],which are expected to provide a theoretical basis for improving goat meat quality through molecular breeding.

Reference information:

  1. Li, L.; Jiang, J.; Wang, L.; Zhong, T.; Chen, B.; Zhan, S.; Zhang, H.; Du, L., Expression patterns of peroxisome proliferator-activated receptor gamma 1 versus gamma 2, and their association with intramuscular fat in goat tissues. Gene 2013, 528, (2), 195-200.
  2. Zhang, M.; Zhang, Z.; Zhang, X.; Lu, C.; Yang, W.; Xie, X.; Xin, H.; Lu, X.; Ni, M.; Yang, X.; Lv, X.; Jiao, P., Effects of dietary Clostridium butyricum and rumen protected fat on meat quality, oxidative stability, and chemical composition of finishing goats. Journal of animal science and biotechnology 2024, 15, (1), 3.
  3. Li, X.; Zhang, H.; Wang, Y.; Li, Y.; Wang, Y.; Zhu, J.; Lin, Y., Chi-Circ_0006511 Positively Regulates the Differentiation of Goat Intramuscular Adipocytes via Novel-miR-87/CD36 Axis. International journal of molecular sciences 2022, 23, (20).

Figure 1: Why do you think miR-196a expression declined from 2mo to 9 mo in LM muscle? Please discuss.

Response2: Thank you for your comments, miR-196a expression declined from 2-month-old to 9-month-old in LM muscle in Fig.1, which was derived from our previous sequencing results [4]. We speculate that 2-month-old to 9-month-old is not the main fat deposition period of goats, which can be verified through IMF content detection of 2-moth, 9-month and 24-month-old goats in our previous study[5]. During this period, miR-196a may not play its role in promoting lipid deposition, so the expression level of miR-196a may decrease. After 9-month-old, goats started to enter the fat deposition period, and the expression level of miR-196a may gradually increase, playing its role in promoting lipid deposition.

Reference information:

  1. Zhang, W.; Liao, Y.; Shao, P.; Yang, Y.; Huang, L.; Du, Z.; Zhang, C.; Wang, Y.; Lin, Y.; Zhu, J., Integrated analysis of differently expressed microRNAs and mRNAs at different postnatal stages reveals intramuscular fat deposition regulation in goats (Capra hircus). Animal genetics 2024.
  2. Lin, Y.; Zhu, J.; Wang, Y.; Li, Q.; Lin, S., Identification of differentially expressed genes through RNA sequencing in goats (Capra hircus) at different postnatal stages. PloS one 2017, 12, (8), e0182602.

Figure 1: Letters in Figure 1 are too small to understand. Please enlarge.

Figure 2: Letters in Figure 2 are too small, most of them are invisible. Please enlarge. In panel C, the background color should be gray and the black/white contrast should be stronger to understand the image.

Figure 3: Letters in the figure are too small, most of them are invisible. Please enlarge. In panel C, the background color should be gray and the black/white contrast should be stronger to understand the image.

Figure 4: Letters in the panels A and B are too small. Please enlarge.

Figure 5: Letters in the panels C-F are too small, most of them are invisible. Please enlarge. In panel E, the background color should be gray and the black/white contrast should be stronger to understand the image. This issue is also true to Figures 6 and 7, needs to be addressed.

Response3: The above questions about the letters are too small to see clearly, we have adjusted the images and enlarged these pictures in Word. Considering that this study involves a lot of pictures, the group of pictures combination may not be so clear in Word. In order to facilitate your understanding, we provide a separate archive to display our pictures. About the oil red O staining image, we have tried to get the white background, but the effect is not satisfying. In contrast, yellow background could clearly display the content of lipid droplets, so we chose yellow background for oil red O pictures. In previous studies [3, 6], Oil red O staining results with yellow background has been selected to show the content of lipid droplets.

Reference information:

  1. Li, X.; Zhang, H.; Wang, Y.; Li, Y.; Wang, Y.; Zhu, J.; Lin, Y., Chi-Circ_0006511 Positively Regulates the Differentiation of Goat Intramuscular Adipocytes via Novel-miR-87/CD36 Axis. International journal of molecular sciences 2022, 23, (20).
  2. Tang, Y.; Zhang, W.; Wang, Y.; Li, H.; Zhang, C.; Wang, Y.; Lin, Y.; Shi, H.; Xiang, H.; Huang, L.; Zhu, J., Expression Variation of CPT1A Induces Lipid Reconstruction in Goat Intramuscular Precursor Adipocytes. International journal of molecular sciences 2023, 24, (17).

Line 257-: Results from miR-196a inhibition by and overexpression quite differed one another. Please discuss it.

Response4: Gene overexpression and interference are common methods to explore gene function. It can elucidate the function of a gene through two sides. In this study, we could find that overexpression of miR-196a promoted adipogenesis and inhibited proliferation of goat intramuscular preadipocytes, while inhibition of miR-196a displayed the opposite results. The overexpression and downregulation results quite differed on another, and both had significant effect in lipid deposition, which not only displayed the key role of miR-196a, but also showed its unique effect that could not be rescue by other molecules. The inhibition effect of miR-196 inhibitor was only 65 percent, and compared with inhibitor, miR-196a mimic had a more significant effect on the phenotype, and there may be a more significant changes at internal molecular level. Therefore, we used miR-196a mimic treatment for RNA-Seq.

Section 3.5: I do not agree to the conclusion that "miR-196a promotes lipid accumulation through activating PI3K-Akt pathway ". Authors examined effect of miR-196a by mimic and effect of LY294002 separately, but not in combination. If they want to conclude in above-mentioned way, they need to perform an examination using mimic and LY294002 in combining way.

Response5: Thank you for your comments. Regarding the conclusion ' miR-196a promotes lipid accumulation by activating the PI3K-Akt pathway ', we found that overexpression of miR-196a increased p-Akt level and p-Akt/Akt ratio (Fig.5A, 5C). LY294002 significantly reduced Akt phosphorylation and p-Akt/Akt ratio (Fig.5B, 5D). In addition, we combined miR-196a mimic and LY294002 treatment in a rescue experiment (Fig.5E-F) using oil red O and Bodipy staining techniques. The above data can explain that miR-196a promotes lipid accumulation by activating the PI3K-Akt pathway to some extent, and we could explore their effect in further studies.

Line 279: Why do authors think miR-196a altered overexpression p-Akt/Akt ratio but did not affect gene expression of adipogenic differentiation nor preadipocyte proliferation in RNA-seq? Please explain.

Response6: Thank you for your question, in line 292 we only described that miR-196a overexpression changed the p-Akt/Akt ratio, indicating that miR-196a may regulate lipid deposition by activating the PI3K-Akt pathway, and did not mention that miR-196a does not affect gene expression of adipogenic differentiation or preadipocyte proliferation in RNA-seq. In RNA-Seq data, genes such as ITGB1 and IGF2 are closely related to adipogenic differentiation and cell proliferation. After treatment with miR-196a mimic, the expression of ITGB1 decreased and the expression of IGF2 increased. ITGB1, also known as integrin β1, is a crucial member of the β subfamily of membrane receptor. ITGB1 is involved in various cellular processes, including cell proliferation, tissue repair, and immune response[7]。ITGB1 is a key gene involved in the regulation of lipid biosynthesis, which is closely related to IMF content [8, 9]. Igf2 plays a central role in gluconeogenic control, lipid metabolism, and insulin signaling pathways and is necessary for maintaining glucose homeostasis as well as the activation of the gluconeogenic regulatory program[10, 11]. Igf2 is reported to inhibit lipid accumulation and inflammatory responses in macrophages[12]. IGF2 is highly mitogenic, and together with IGF1, it promotes the proliferation of various types of cells during the fetal period, thereby playing a major role in organ growth and development[10].

Reference information:

  1. Qi, R.; Hou, J.; Yang, Y.; Yang, Z.; Wu, L.; Qiao, T.; Wang, X.; Song, D., Integrin beta1 mediates the effect of telocytes on mesenchymal stem cell proliferation and migration in the treatment of acute lung injury. Journal of cellular and molecular medicine 2023, 27, (24), 3980-3994.
  2. Yu, H.; Yu, S.; Guo, J.; Wang, J.; Mei, C.; Abbas Raza, S. H.; Cheng, G.; Zan, L., Comprehensive Analysis of Transcriptome and Metabolome Reveals Regulatory Mechanism of Intramuscular Fat Content in Beef Cattle. Journal of agricultural and food chemistry 2024, 72, (6), 2911-2924.
  3. Fang, Y.; Wu, Y.; Liu, L.; Wang, H., The Four Key Genes Participated in and Maintained Atrial Fibrillation Process via Reprogramming Lipid Metabolism in AF Patients. Frontiers in genetics 2022, 13, 821754.
  4. Sélénou, C.; Brioude, F.; Giabicani, E.; Sobrier, M. L.; Netchine, I., IGF2: Development, Genetic and Epigenetic Abnormalities. Cells 2022, 11, (12).
  5. Halmos, T.; Suba, I., [The physiological role of growth hormone and insulin-like growth factors]. Orvosi hetilap 2019, 160, (45), 1774-1783.
  6. Qiao, X. R.; Wang, L.; Liu, M.; Tian, Y.; Chen, T., MiR-210-3p attenuates lipid accumulation and inflammation in atherosclerosis by repressing IGF2. Bioscience, biotechnology, and biochemistry 2020, 84, (2), 321-329.

We sincerely appreciate for your comments, and hope the responses will clarify the problems.

Best regards

Reviewer 3 Report

Comments and Suggestions for Authors

 In manuscript written by Yuling Yang  , Wenyang Zhang  , Haiyang Li  , Hua Xiang, Changhui Zhang  , Zhanyu Du  , Lian Huang and Jiangjiang Zhu,  based on previously described notion   that miRNA plays a crucial role in adipocyte proliferation and differentiation, observed the expression variations of miR-196a in the longissimus dorsi  muscle of Jianzhou goats at different ages. The results of this study showed that miR-196a could promote adipogenesis, apoptosis and inhibit cell proliferation in goat intramuscular preadipocytes. Then, through RNA-seq and in vitro verification, authors   found that miR-196a regulated intramuscular preadipocytes adipogenesis through activating PI3K-Akt signaling pathway.  Authors utilized RT-qPCR, western blot, dual-luciferase and rescue assays.  Authors concluded that miR-196a could promote adipogenesis, apoptosis and inhibit cell proliferation in goat intramuscular preadipocytes.  Authors described that miR-196a directly targets MAP3K1, which was closely related to adipogenesis, to facilitate lipid deposition in preadipocytes. In conclusions, these data clarify that miR-196a can control lipid formation through targeting MAP3K1 and activating PI3K- Akt signaling pathway and provide the experimental basis for delineating the regulatory network of miR-196a in regulating goat IMF content. The current manuscript is written well, detailed description of assays used in studies are presented clearly   and   convincible in a paper,  conclusions based on results are valid.

 I have some specific comments:

1.Figure 3G presents results of RT-PCR  for relative expression of apoptosis-related genes after interfering with miR-196a.   I suggest confirming these data by others methods  such as western blot to show   increased protein expression of caspases and activation  of  caspases. Activated caspases or cleaved PARP are the best indicators for developing of apoptosis. Figure 3F shows activation of apoptosis by FLOW for annexin V -FITC expression as an indicator of apoptosis but additional line of evidence of activated caspases and PARP will make those conclusions stronger.

2. I found interesting observations presented in Figure 4c: the top 30 enriched pathways of DEGs after miR-196a overexpression in preadipocytes. One of the most enriched pathways is Herpes simplex virus 1 infection following by PI3K-Akt signaling pathways. There are a lot of literature data indicated the HSV-1and HSV-2 infections activates PI3K and Akt signaling pathways in different cells including primary cells. Activation of Akt could be because of preexisting infection of HSV-1. Also, HSV infection upregulates expression of some miRNAs. Can authors comment on it?

Figure 6E. Western blots show expression of MAP3K1 in cells transfected with miR-196a mimic, miR-196a inhibitor or their corresponding NCs. The quality of presented western blots is not good (too narrow bands). Would authors present western blots with better quality? The same for Figure 7A.  

Author Response

Reply to the Review Report

Dear reviewer,

Thanks for your comments concerning our manuscript entitled “MiR-196a Promotes Lipid Deposition in Goat Intramuscular Preadipocytes by Targeting MAP3K1 and Activating PI3K-Akt Pathway” (cells-3114823). Those comments were valuable and very helpful. We have read the comments and the instructions for authors carefully. Revised portions are marked in yellow in the revised manuscript.

Comments and Suggestions for Authors

In manuscript written by Yuling Yang  , Wenyang Zhang  , Haiyang Li  , Hua Xiang, Changhui Zhang  , Zhanyu Du  , Lian Huang and Jiangjiang Zhu,  based on previously described notion   that miRNA plays a crucial role in adipocyte proliferation and differentiation, observed the expression variations of miR-196a in the longissimus dorsi  muscle of Jianzhou goats at different ages. The results of this study showed that miR-196a could promote adipogenesis, apoptosis and inhibit cell proliferation in goat intramuscular preadipocytes. Then, through RNA-seq and in vitro verification, authors   found that miR-196a regulated intramuscular preadipocytes adipogenesis through activating PI3K-Akt signaling pathway.  Authors utilized RT-qPCR, western blot, dual-luciferase and rescue assays.  Authors concluded that miR-196a could promote adipogenesis, apoptosis and inhibit cell proliferation in goat intramuscular preadipocytes.  Authors described that miR-196a directly targets MAP3K1, which was closely related to adipogenesis, to facilitate lipid deposition in preadipocytes. In conclusions, these data clarify that miR-196a can control lipid formation through targeting MAP3K1 and activating PI3K- Akt signaling pathway and provide the experimental basis for delineating the regulatory network of miR-196a in regulating goat IMF content. The current manuscript is written well, detailed description of assays used in studies are presented clearly   and   convincible in a paper, conclusions based on results are valid.

1.Figure 3G presents results of RT-PCR for relative expression of apoptosis-related genes after interfering with miR-196a.   I suggest confirming these data by others methods such as western blot to show   increased protein expression of caspases and activation of caspases. Activated caspases or cleaved PARP are the best indicators for developing of apoptosis. Figure 3F shows activation of apoptosis by FLOW for annexin V -FITC expression as an indicator of apoptosis but additional line of evidence of activated caspases and PARP will make those conclusions stronger.

Response 1: We agree with you that western blot can make our conclusion more effective. Therefore, we supplemented the protein expression of Caspase3 after overexpression and interference with miR-196a in Fig.2H and Fig.3H, respectively. Caspase3 protein expression increased after overexpression of miR-196a and decreased after interference with miR-196a, which verified our conclusion. Unfortunately, since the laboratory does not have a suitable PARP antibody, we are temporarily unable to provide Western blotting results on PARP. I believe that the WB results of Caspase3 have supported the conclusion that miR-196a promotes apoptosis to some extent. The result description of the corresponding supplementary experiment has been added to the manuscript (highlighted).

  1. I found interesting observations presented in Figure 4c: the top 30 enriched pathways of DEGs after miR-196a overexpression in preadipocytes. One of the most enriched pathways is Herpes simplex virus 1 infection following by PI3K-Akt signaling pathways. There are a lot of literature data indicated the HSV-1and HSV-2 infections activates PI3K and Akt signaling pathways in different cells including primary cells. Activation of Akt could be because of preexisting infection of HSV-1. Also, HSV infection upregulates expression of some miRNAs. Can authors comment on it?

Response 2: Thanks for your comments, we added the following discussion in the manuscript line 424-438: In Fig.4C, we observed that Herpes simplex virus 1 infection pathway was significantly enriched,Herpes simplex virus type 1 (HSV-1) and type 2 (HSV-2) are enveloped double stranded DNA viruses belonging to Herpesviridae [1]. PI3K-Akt signaling pathway is essential for cell survival and can be activated by various viruses such as HIV-1 and HSV-1 [2, 3]. HSV-1 infection increased the expression of P-PI3K, P-AKT and p-p38 in Hela and Vero cells [3]. Moreover, sophoridine (SRI) treatment significantly inhibited HSV-1 infection-induced phosphorylation of PI3K, Akt and mTOR in Vero and HeLa cells [4]. A total of 27 genes were enriched in the Herpes simplex virus 1 infection pathway in the RNA-Seq data of this study. Among them, component 3 (C3) has a protective effect on delaying keratitis and neovascularization after HSV-1 infection [5]. HSV-1 Susceptibility Increases in Fibroblasts of STAT1 Deficient Patients [6]. Therefore, we speculate that miR-196a may promote HSV-1 infection by down-regulating the expression of genes such as C3 and STAT1, and then activate the PI3K-Akt pathway. The specific molecular mechanisms involved need to be further studied.

Reference information:

  1. Fatahzadeh, M.; Schwartz, R. A., Human herpes simplex virus infections: epidemiology, pathogenesis, symptomatology, diagnosis, and management. Journal of the American Academy of Dermatology 2007, 57, (5), 737-63; quiz 764-6.
  2. Pardons, M.; Cole, B.; Lambrechts, L.; van Snippenberg, W.; Rutsaert, S.; Noppe, Y.; De Langhe, N.; Dhondt, A.; Vega, J.; Eyassu, F.; Nijs, E.; Van Gulck, E.; Boden, D.; Vandekerckhove, L., Potent latency reversal by Tat RNA-containing nanoparticle enables multi-omic analysis of the HIV-1 reservoir. Nature communications 2023, 14, (1), 8397.
  3. Liu, Y.; Chen, L.; Liu, W.; Li, D.; Zeng, J.; Tang, Q.; Zhang, Y.; Luan, F.; Zeng, N., Cepharanthine Suppresses Herpes Simplex Virus Type 1 Replication Through the Downregulation of the PI3K/Akt and p38 MAPK Signaling Pathways. Frontiers in microbiology 2021, 12, 795756.
  4. Tang, Q.; Luan, F.; Yuan, A.; Sun, J.; Rao, Z.; Wang, B.; Liu, Y.; Zeng, N., Sophoridine Suppresses Herpes Simplex Virus Type 1 Infection by Blocking the Activation of Cellular PI3K/Akt and p38 MAPK Pathways. Frontiers in microbiology 2022, 13, 872505.
  5. Filiberti, A.; Gmyrek, G. B.; Berube, A. N.; Royer, D. J.; Carr, D. J. J., An intact complement system dampens cornea inflammation during acute primary HSV-1 infection. Scientific reports 2021, 11, (1), 10247.
  6. Lafaille, F. G.; Harschnitz, O.; Lee, Y. S.; Zhang, P.; Hasek, M. L.; Kerner, G.; Itan, Y.; Ewaleifoh, O.; Rapaport, F.; Carlile, T. M.; Carter-Timofte, M. E.; Paquet, D.; Dobbs, K.; Zimmer, B.; Gao, D.; Rojas-Duran, M. F.; Kwart, D.; Rattina, V.; Ciancanelli, M. J.; McAlpine, J. L.; Lorenzo, L.; Boucherit, S.; Rozenberg, F.; Halwani, R.; Henry, B.; Amenzoui, N.; Alsum, Z.; Marques, L.; Church, J. A.; Al-Muhsen, S.; Tardieu, M.; Bousfiha, A. A.; Paludan, S. R.; Mogensen, T. H.; Quintana-Murci, L.; Tessier-Lavigne, M.; Smith, G. A.; Notarangelo, L. D.; Studer, L.; Gilbert, W.; Abel, L.; Casanova, J. L.; Zhang, S. Y., Human SNORA31 variations impair cortical neuron-intrinsic immunity to HSV-1 and underlie herpes simplex encephalitis. Nature medicine 2019, 25, (12), 1873-1884.

Figure 6E. Western blots show expression of MAP3K1 in cells transfected with miR-196a mimic, miR-196a inhibitor or their corresponding NCs. The quality of presented western blots is not good (too narrow bands). Would authors present western blots with better quality? The same for Figure 7A.  

Response 3: We fully agree with your opinion, and we also acknowledge the poor image quality of western blots on MAP3K1. Since the species involved in this study is goat, it is not easy to find suitable antibodies. After trying multiple antibodies, we found that only the antibody produced by Bioss (bs-18780R) can be used and the effect is not good. Unfortunately, we are unable to provide better image of western blots due to antibody factors. In the future, we will further explore this aspect in order to present high-quality western blots images.

We sincerely appreciate for your comments, and hope the responses will clarify the problems.

Best regards

Round 2

Reviewer 1 Report

Comments and Suggestions for Authors

The revised version of manuscript by Yuling Yang and coworkers, entitled: “MiR-196a Promotes Lipid Deposition in Goat Intramuscular Preadipocytes by Targeting MAP3K1 and Activating PI3K-Akt Pathway” has been improved and partially corrected. The Authors took into consideration my remarks and corrected the main body text, as well as the figure captions. I am disappointed, however; that the Authors did not improve the figures at all, and left them unchanged. Leaving the images of goat adipocytes presented at such a low magnification (100x, which means 10x objective) does not allow the reader to observe cellular morphology. The Authors presented images taken at higher magnification in the letter responding to my comments, but they did not replace the images in the manuscript.  I also maintain my doubts regarding the effect of treatment with LY294002. The Authors did not choose to replace the WB images in Figure 5B, with more representative images showing more clearly the inhibitory effect of LY294002 on Akt phosphorylation, but maintained the old version of Figure 5. Nevertheless, I appreciate all changes that have been made, the presented results, the experimental design and methodology. Therefore, I leave the final decision regarding the revised version of the manuscript to the Editors.

Author Response

Reply to the Review Report

Dear reviewer,

Thanks for your comments concerning our manuscript entitled “MiR-196a Promotes Lipid Deposition in Goat Intramuscular Preadipocytes by Targeting MAP3K1 and Activating PI3K-Akt Pathway” (cells-3114823). Those comments were valuable and very helpful. We have read the comments and the instructions for authors carefully. Revised portions are marked in yellow in the revised manuscript.

Comments and Suggestions for Authors

The revised version of manuscript by Yuling Yang and coworkers, entitled: “MiR-196a Promotes Lipid Deposition in Goat Intramuscular Preadipocytes by Targeting MAP3K1 and Activating PI3K-Akt Pathway” has been improved and partially corrected. The Authors took into consideration my remarks and corrected the main body text, as well as the figure captions. I am disappointed, however; that the Authors did not improve the figures at all, and left them unchanged. Leaving the images of goat adipocytes presented at such a low magnification (100x, which means 10x objective) does not allow the reader to observe cellular morphology. The Authors presented images taken at higher magnification in the letter responding to my comments, but they did not replace the images in the manuscript.  I also maintain my doubts regarding the effect of treatment with LY294002. The Authors did not choose to replace the WB images in Figure 5B, with more representative images showing more clearly the inhibitory effect of LY294002 on Akt phosphorylation, but maintained the old version of Figure 5. Nevertheless, I appreciate all changes that have been made, the presented results, the experimental design and methodology. Therefore, I leave the final decision regarding the revised version of the manuscript to the Editors.

Response: We sincerely appreciate your feedback, which will help improve the quality of our manuscript. Regarding the Oil red O images, we acknowledge that the magnification of 100X cannot clearly show cell morphology, but please forgive us for not being able to supplement a large number of experimental results within 5 days, which may take up to a month to make up. Taking into account your suggestions, we have adjusted the Oil red O images and partially enlarged Fig. 2C, 3C, 5E, 7C and E so that readers can better observe the changes in lipid droplet content. We believe that the adjusted Oil red O images will not affect readers' reading and understanding of this study. In addition, we really agree with your comments, and we will improve our photo-taking methods in future studies.

For the effect of LY294002 treatment, we took your opinion and used more representative WB images to more clearly show the inhibitory effect of LY294002 on Akt phosphorylation. The WB images in the manuscript Fig. 5B has been replaced.

Finally, I sincerely thank you for your recognition of the changes we have made in the presented results, the experimental design and methodology.

We sincerely appreciate for your comments, and hope the responses will clarify the problems.

Best regards

Reviewer 2 Report

Comments and Suggestions for Authors

The authors appropriately addressed to Reviewer's comments in the revised manuscript. I agree to the acceptance for publication in the current form.

Author Response

Reply to the Review Report

Dear reviewer,

Thanks for your comments concerning our manuscript entitled “MiR-196a Promotes Lipid Deposition in Goat Intramuscular Preadipocytes by Targeting MAP3K1 and Activating PI3K-Akt Pathway” (cells-3114823).

Comments and Suggestions for Authors

The authors appropriately addressed to Reviewer's comments in the revised manuscript. I agree to the acceptance for publication in the current form.

Response: We sincerely appreciate the time and effort invested by the reviewers in evaluating our manuscript. Thank you very much for your careful review and thoughtful evaluation of my paper. Finally, thank you for recognizing our work.

We sincerely appreciate for your comments.

Best regards
